# Complex early childhood experiences: Characteristics of Northern Territory children across health, education and child protection data

Lucinda Roper[1,2]*, Vincent Yaofeng He[2], Oscar Perez-Concha[1], Steven Guthridge[2]

**1** Centre for Big Data Research in Health, University of New South Wales, Sydney, Australia, **2** Centre for Child Development and Education, Menzies School of Health Research, Charles Darwin University, Darwin, Australia

* roper.lucinda@gmail.com

**Data Availability Statement:** The study datasets contain sensitive personal information and are held on a secure cloud-based server with restricted access. Access requires the approval of the ethics

## Abstract

Early identification of vulnerable children to protect them from harm and support them in achieving their long-term potential is a community priority. This is particularly important in the Northern Territory (NT) of Australia, where Aboriginal children are about 40% of all children, and for whom the trauma and disadvantage experienced by Aboriginal Australians has ongoing intergenerational impacts. Given that shared social determinants influence child outcomes across the domains of health, education and welfare, there is growing interest in collaborative interventions that simultaneously respond to outcomes in all domains. There is increasing recognition that many children receive services from multiple NT government agencies, however there is limited understanding of the pattern and scale of overlap of these services. In this paper, NT health, education, child protection and perinatal datasets have been linked for the first time. The records of 8,267 children born in the NT in 2006–2009 were analysed using a person-centred analytic approach. Unsupervised machine learning techniques were used to discover clusters of NT children who experience different patterns of risk. Modelling revealed four or five distinct clusters including a cluster of children who are predominantly ill and experience some neglect, a cluster who predominantly experience abuse and a cluster who predominantly experience neglect. These three, high risk clusters all have low school attendance and together comprise 10–15% of the population. There is a large group of thriving children, with low health needs, high school attendance and low CPS contact. Finally, an unexpected cluster is a modestly sized group of non-attendees, mostly Aboriginal children, who have low school attendance but are otherwise thriving. The high risk groups experience vulnerability in all three domains of health, education and child protection, supporting the need for a flexible, rather than strictly differentiated response. Interagency cooperation would be valuable to provide a suitably collective and coordinated response for the most vulnerable children.

committee and data custodians. For applications for data access, please contact the Menzies Data-linkage Program Leader at steve.guthridge@menzies.edu.au, or the Menzies Data Management Team at datamanagement@menzies.edu.au.

**Funding:** The study was completed by LR as a component of her post-graduate studies. The Child and Youth Development Research Partnership (CYDRP) data repository is supported by a grant from the NT Government. The funders of the repository had no role in study design, data preparation, analysis, decision to publish or preparation of the manuscript.

**Competing interests:** The authors have declared that no competing interests exist.

# Background

As a society we aim to support the optimal development of children. Social and economic factors, such as parental education, income, drug and alcohol use or housing shape the environments that children are exposed to, which may support or hinder their development [1]. These environments effect development in all domains, including physical, mental and emotional health as well as educational attainment. The same biopsychosocial factors which can detriment a child's health can also cause unstable and potentially unsafe home environments, which can result in family contact with the child protection system. For instance, poverty and crowding are associated with higher levels of respiratory and ear infections and are also associated with contact with the child protection service [2, 3]. Poor physical health and contact with child protection services are both associated with reduced educational attainment [4–6]. Early life experiences have a long term effect on later health and function, as key social, emotional, cognitive and physical capabilities are learnt at this stage [7–9]. Early life intervention (generally accepted as the first 8 years of life) is therefore a recognised priority area. This can include intervening when societal expectations are not met–for instance in response to low school attendance, child maltreatment or poor health.

## Data from different agencies to support the needs of vulnerable children

Australia, like all developed countries, has multiple government and non-government organisations that support children by identifying families at need and providing services. Specifically, this occurs across the domains of health, education and child protection. Data from these agencies is not only useful to record services for administrative reasons, but can enable research to inform responses, particularly when data can be shared and linked across agencies [10]. This is because there is increasing recognition that interagency collaboration is essential to holistically support these children, given the shared dimensions of health that influence outcomes across these areas [11–13]. Linked, population-wide, administrative health, education and social service data are increasingly recognised as powerful tools to provide insight into factors affecting childhood development, to improve services and interventions [10].

## Administrative datasets available in Australia for study of child development

Child protection data typically includes observational data of children who have contact with the child protection service and has been used to predict a range of developmental outcomes, including physiological outcomes, emotional functioning, cognitive and academic performance and also healthcare service utilisation [14–21]. More recently, population level, administrative child protection data has been used to build predictive models to identify children at highest risk of harm–and offer those families early intervention before the predicted harm has occurred [22]. There are however ethical concerns surrounding the use of predictive models in social services, particularly relating to privacy, stigma and 'self-fulfilling prophecy' for families identified as high risk [23].

Health data used in the study of vulnerable children typically measures the impact of disease on early childhood development [24–26], via data on hospital admissions in the early years or using perinatal data to extract birth risks. Chronic conditions have been the most extensively studied, and a diagnosis of any chronic condition is associated with poorer educational outcomes at a range of ages [27–31], with comparable findings in studies of specific conditions such as asthma [25, 30], cerebral palsy [26] or epilepsy [24]. Similar effects have been found for

childhood infections [32, 33], injuries [34–36], and overall number of ED presentations and total hospital inpatient days between birth and age seven [5].

Given the impact of childhood illnesses on development, an alternative paradigm is consideration of the preventability of these conditions—many illnesses are associated with poverty and failure to receive preventative care or timely care (e.g. vaccination, dental care, asthma management). The Australian Institute of Health and Welfare (AIHW) list of ambulatory care sensitive conditions (ACSC) has been used for pediatric research [37], but misses numerous common pediatric conditions, such as gastroenteritis [38]. Several pediatric-specific indicators of avoidable hospitalization have been developed in the United States, United Kingdom and New Zealand (NZ) [39, 40]. The NZ indicator suite includes admissions that may be avoidable via policy measures that influence the socioeconomic determinants of health [41]. This allows a broader range of conditions to be captured and prevents unrealistic expectations of the role of healthcare in prevention, when larger social policies affecting socioeconomic gradients also play a role. The NZ indicator suite has been used in Australia to compare avoidable hospitalizations in New South Wales Aboriginal and non-Aboriginal children [41, 42]. The pediatric indicator found more avoidable hospitalizations than the adult indicator, and these hospitalizations were more associated with social and health disadvantage at birth [38].

The Australian education system collects data on enrolment and attendance, and also performance on nationally standardised tests at years 3, 5, 7 and 9 (National Assessment Program–Literacy and Numeracy, NAPLAN) [43]. More recently, Australia has also begun collecting information through the Australian Early Development Census (AEDC) [44]–a three-yearly nationwide data collection of early childhood development, completed when children are in their first year of full-time school. Education data is used by government to monitor performance against national targets, and by educators to assess the causal impact of a wide range of factors on performance (i.e. pre-school attendance, teacher qualifications, school funding model) [45, 46]. Furthermore, this data has been used by human development economists to understand how schooling relates to later outcomes–e.g. correlations with school dropout, delinquency, and involvement with the legal system [47, 48]. Australian studies indicate that early variation in attendance rates persist over the school career [49]. Whilst there has traditionally been an assumption that attendance and performance are linked, it is not clear how policies aimed at improving attendance affect performance, particularly for Aboriginal schoolchildren [50].

## Limitations of current studies

Administrative Health, Education and Child Protection Services (CPS) data have rarely been linked together in Australia; datasets are typically considered either individually or in pairs. For instance, linked hospital and CPS data have been analysed in New South Wales (NSW) and Western Australia (WA), and it has been established that maltreated children have higher rates of healthcare utilisation [51–53]. Research using these data sources together has focused on improving identification of children at risk of harm [51, 54] or defining healthcare costs to build an economic case for policy change [17, 52, 55–57]. Linked hospital and education data have also been studied together, to clearly establish that illness has a detrimental effect on educational attainment [24, 25, 33]. CPS and educational data have been studied together, in the NT and elsewhere [58], with many studies focusing on the effect of specific elements of child abuse on educational outcomes. International studies indicate that adverse outcomes such as school dropout, are related to whether multiple types of abuse occurred and the chronicity and severity of maltreatment [14, 15, 18–21, 59–62].

Not only has previous consideration of these datasets been limited to pairs; frequently one dataset is assumed to represent the outcome–for instance hospitalisation as an outcome of child maltreatment. However, increased hospital utilisation and CPS involvement may sometimes have a reverse relationship: an ill child may cause household stress and increased propensity to maltreatment, as well as increased exposure to health professionals, who are the main CPS reporter group in early childhood [63]. Therefore, although there has been research into vulnerable children in one or two dimensions, their overlapping risks across these three domains are not fully understood. There may be subpopulations with different patterns of risk, who may benefit from differentiated responses.

## Context of the current study

The Northern Territory (NT) of Australia is the smallest of Australia's states and territories by population, comprising approximately 1% of the total Australian population [64]. The NT is characterised by its large Aboriginal population, comprising approximately 30% of the NT population, compared to the national average of 3% of the population [65]. The NT Aboriginal population has a relatively young age structure, with a median age of 26 years compared to 35 years for the non-Aboriginal population [64] and Aboriginal children make up 43% of all NT children [66].

Aboriginal Australians experience disadvantage in almost all measures of health and welfare, and in the NT, the Aboriginal population experience disproportionate levels of poverty, crowded housing and poor health [67–70]. Although Aboriginal children make up 43% of NT children, they comprise 82% of children in contact with the CPS [71] and also have higher rates of avoidable hospitalisations [42, 72] and lower rates of school attendance [49, 73]. School attendance, particularly in the early years, allows children to develop a foundation for later education and learning. NT Aboriginal students have average attendance rates below 60% [74], whereas the Australian Curriculum, Assessment and Reporting Authority uses 90% attendance rate as a key performance measure to assess adequate attendance level [75].

In Australia, the suite of government research and policies aimed at reducing inequality between Aboriginal and non-Aboriginal Australians are known as "Closing the Gap" [76]. Initially referring to closing the life expectancy gap, this term also applies to gaps in privilege and attainment in a wide variety of areas–including health, education and employment. By 2018, the 2014 target: to close the gap between Aboriginal and non-Aboriginal students' attendance rates, within 5 years, had not been met [77]. The 2020 updated National Agreement on Closing the Gap contains 17 national targets, three of which are specifically relevant to this project:

- Aboriginal and Torres Strait Islander children are not overrepresented in the child protection system

- Aboriginal and Torres Strait Islander children thrive in their early year, and

- Aboriginal and Torres Strait Islander children are engaged in high quality, culturally appropriate early childhood education in their early years

In the NT, government agencies involved in the identification of vulnerable children include the NT Department of Health (Health), NT Department of Education (Education) and the Department of Territory Families, Housing and Communities (child protection service (CPS)). Each has their own data systems and policies for sharing and management of data. There is currently limited understanding of how children are treated by or notified to multiple services. However, there is increasing recognition that children may be seen by all three NT government agencies and that a multi-agency person-centred response could be beneficial

[78–80]. Thus, the NT Government has collaborated with the Menzies School of Health Research's Centre for Child Development and Education (CCDE) and invested in the Child and Youth Development Research Partnership (CYDRP). This collaborative research partnership between CCDE and the NT Government supports the ongoing maintenance and development of an extensive data repository to be used for approved research projects that inform the health and wellbeing of NT children

The motivation for this present study was to discover if routinely collected data on NT children can be used to separate children into groups who may benefit from differentiated responses, including collaborative inter-agency interventions for the most vulnerable. To remain sensitive to the complexity of the relationships between different indicators of disadvantage it was considered that an exploratory, data-driven approach, such as clustering, was most likely to return useful information, irrespective of causality, to inform planning for multi-agency approaches.

## Methods

### Data sources

This project utilises four datasets from the Northern Territory (NT)–the Perinatal Data Register, Hospital Admissions, School Enrolment and Attendance and Child Protection Service (CPS) contacts. The data are held in a data repository containing de-identified, unit-level linked records for NT children across a total of 14 datasets. The repository is longitudinal dataset, developed through the Child and Youth Development Research Partnership (CYDRP) between Menzies School of Health Research and multiple NT Government agencies, including the departments of Health, Education, and Territory Families, Housing and Communities (child protection) [81]. Initial linkage was conducted by SA NT Datalink [82], using probabilistic methods to match the records for children across multiple datasets, with clerical review of uncertain matches. This process is confirmed to result in 99.6% accuracy for completed links [82]. SA NT DataLink creates a unique linkage key for each child and provides this to data custodians. Each data custodian then creates a de-identified research dataset containing only the linkage keys and approved research variables, which is provided to researchers. Researchers are then able to merge records for the same child across multiple datasets. A full description is available elsewhere [83].

The NT Perinatal Data Register (perinatal data) is a statutory collection containing demographic, antenatal and birth information for all births in the NT and was used to define the cohort. The Hospital Admissions dataset (hospital data) contains information on all admissions to public hospitals in the NT. The Education Enrolment and Attendance dataset (education data) contains information on daily school attendance for children attending NT Government schools. Approximately 70% of all NT children attended government schools in 2020. Nationally, 65% of all Australian children and 83% of Aboriginal children attended government schools [84]. The CPS dataset contains information on all notifications of suspected maltreatment, with further details for those notifications that are investigated and substantiated.

### Study population

The study population comprised all children born in the NT between 2006–2009 and present in the NT Government school attendance records in Year 1 (usually age 5–6). Of the 15,284 children born during the selection period, 8,267 children were retained in the study after excluding those with no records in NT Government school attendance records in Year 1. Children born to mothers who were resident of the major centres of Darwin and Alice Springs and their hinterlands at time of birth were classified as urban residents; all other children were classified as remote residents.

### Research question

Can we discover distinct groups of NT children, at the end of their first year of formal schooling, with varying patterns of contact with the CPS, Health and Education services?

### Analysis approach

1. <u>Initial descriptive analysis</u>

Initial descriptive analysis was undertaken, including counts and percentages for categorical variables and mean, median and standard deviation for continuous variables, and data visualisations including histograms and bar plots, to provide an overview of the data. The results were stratified by Aboriginal status, as disparities in the health, social and educational outcomes between Aboriginal and non-Aboriginal children in the Northern Territory were expected.

### Health data

Three components of the hospital data were explored:

- Any hospitalisation (binary) from birth to age 5: overall and with a specific diagnosis. The specific diagnoses explored were 'avoidable hospitalisations'–defined as a diagnosis on the Anderson avoidable hospitalisation list [56]. This is a list of 26 conditions represented by 130 ICD-10-AM codes, which can be found in the appendix of the original paper [85]. The birth admission was excluded.

- Number of hospitalisations (count) from birth to age 5: overall or with a specific diagnosis (diagnosis codes defined as per above).

- Length of stay per hospitalisation (average per child)

### Child protection service data

Five child protection variables were explored. For each variable, both binary (any experience of outcome) and count (number) were explored. In the NT there is a statutory requirement that requires all adults to report any incident, to either the Department of Territory Families Housing and Communities, or the police, if they have a reasonable belief that a child has been or is likely to be harmed [86, 87]. S1 Appendix contains further detail on rationale for inclusion of each child protection variable into the models.

- Notifications: a notification is a report made to the CPS by a reporter that a child is experiencing or at risk of neglect or abuse.

- Substantiations: after a notification, some notifications are investigated, based on a risk assessment. Approximately one third of investigated notifications result in substantiations (i.e. confirmed abuse or neglect).

- Abuse type: physical, emotional, sexual abuse or neglect. A description of physical, emotional, sexual abuse or neglect can be found here [87]. Emotional harm occurs when a child's social, emotional or cognitive development is impaired or is at significant risk as a result of their parents' or carers' persistent failure to meet the child's emotional need for love and security, or their psychological needs for stimulation and nurturing. This includes exposure to family or domestic violence, which requires an automatic report for emotional abuse.

- Reporter type: this is the recorded occupation of the reporter. This has been grouped into eight categories: community members (including family or self-reporting), child protection officer, school personnel, police, health professional, non-government organisation, other professional (including social workers) and not stated.

- Substantiation descriptor: if a notification is substantiated, then an additional categorical variable–the substantiation descriptor–is recorded. Although there are many different substantiation descriptors, alcohol and other drugs (AOD) or domestic violence (DV) are the most common and are explored here.

### Educational data

The educational engagement measure chosen was Year 1 school attendance, defined as total days attended in Year 1, divided by total days expected to attend. Expected to attend is a variable provided by school data to represent ideal attendance for that specific child, i.e. a child who moved schools and was only enrolled for 6 months, is only expected to attend for 6 months

2. Data-driven approach: clustering

In our context, a cluster is defined as a collection of children aggregated together because of similarities in regard to the variables used in this study, using a distance-based similarity measure, such as Euclidian Distance. We used a data-driven approach, in which representations of distinct groups of NT children are automatically learned using unsupervised machine learning techniques. Unsupervised learning, such as clustering, uses data that is unlabelled, meaning that children have not been pre-assigned any labels or categories by the researcher. Instead of using a pre-defined outcome to form groups, the algorithm discovers patterns in the data using only the input data [88]. The k-means clustering methodology was used.

It is common practice to stratify analysis by Aboriginal status, because of the major differences in a range of conditions between Aboriginal and non-Aboriginal populations. In this study, however, we did not split the cohort by Aboriginal status for the clustering, for two reasons. Firstly, keeping both groups allowed assessment of the plausibility of the clusters. Given known differences in health, education, and social outcomes between Aboriginal and non-Aboriginal children, we would expect most non-Aboriginal children to be in a relatively healthy cluster. Secondly, because there are no requirements for the clusters to be the same size, a small, highly vulnerable cluster can be discernible, irrespective of the number of low-risk children in the cohort. Unlike traditional modelling which may struggle to estimate parameters for both groups, clustering is very flexible and can accommodate diverse samples.

### Clustering methodology: K-means

K-means is a simple and efficient algorithm, popular in health research [89], a full explanation can be found in S2 Appendix. In summary, it is a distance-based algorithm (Euclidian Distance), which follows the below steps:

a.  Pre-select the number of clusters

b.  Initiation step: several cluster centres are randomly chosen

c.  Assignment/Expectation step: the Euclidean distance between each datapoint and each initial cluster centre is calculated and each datapoint is assigned to the cluster it is nearest to

d.  Update/Maximisation step: cluster centres are updated to be the mean of all points that were assigned to that cluster

e. The algorithm continues to move between steps c and d until convergence is reached (when the assignment of points to a cluster no longer changes)

The 'ideal' number of clusters will result in the most compact and separated clusters. We used the silhouette coefficient to assess this, which is explained in the supplementary material. It is a score that captures dissimilarity between clusters and similarity within clusters [90].

## Dimensionality reduction

K-means cannot deal with the 'curse of dimensionality', meaning that its performance can suffer in high-dimensionality datasets. With increasing dimensions, each observation in the dataset appears similarly distant to all others as the more dimensions involved, the greater chance that a difference apparent in one dimensions becomes nullified by similarity in another [91, 92]. A more detailed explanation of this is in S3 Appendix.

There has therefore been substantial work done on variable subset selection for use in clustering [93–97]. It is a challenging problem, firstly, because there are no labels to allow for evaluation of variable importance based on classification accuracy (supervised machine learning) or univariate relationship with the outcome. Secondly, because the number of clusters isn't predetermined, and the cluster number affects variable importance, it is difficult to unpick the entwined issues of cluster number and variable selection [96].

We have taken the following approach:

a. Dimensionality reduction, via three approaches outlined below

b. Clustering on reduced dimensionality data

c. Map the reduced dimensionality data, with their cluster assignments, to the full dataset, to characterise the clusters

We therefore created three separate cluster structures, by using three separate approaches to dimensionality reduction. These are explained in detail in the supplementary material. Firstly, we used two well-established pre-clustering variable selection methods: filtering and feature extraction via principal component analysis. Secondly, we combined these with a more novel approach, which iteratively uses post-clustering variable importance rankings, to define a new variable subset.

- Method 1: Expert selection: the use of expert knowledge, guided by the principle of diversity, to choose a subset of variables which represent unique aspects of the data whilst having a presumed relationship to the outcome of interest. This is also known as a filter method [94, 97].

- Method 2: PCA: PCA is a method of automatic dimensionality reduction that projects the original data into a smaller number of dimensions [98]. PCA was carried out using the scikit-learn package, set to reduce the data to 4 dimensions [90].

- Method 3: post-clustering variable extraction via decision trees, following the steps below:

  a. create clusters based on the variable subset resulting from method 1 (features extracted by human expert) and method 2 (features extracted by PCA)

  b. Predict cluster membership using the Extra Tree Classifier in scikit-learn [90] and rank all original 43 variables based on their contribution to predicting cluster membership–for details on this variable importance metric, see [99]

  c. Select the top 10 ranked variables from each model and use PCA to reduce these to four dimensions

### Post hoc analysis–mapping clusters done in reduced dimension to original data

Cluster labels were assigned to each data point, and a descriptive analysis was performed, within clusters, for variables within the health, CPS, education and perinatal datasets, with additional demographic information relevant to the NT. Informative names were then assigned to each cluster based on patterns revealed in the descriptive analysis.

Given that each cluster was generated in a different, reduced-dimensionality subspace, there is no objective method to assess which method generated the 'best' clusters, as we cannot compare their silhouette scores. We have therefore been informed by the concept of evidence accumulation [100]. This was originally a method to combine multiple cluster structures within a process known as consensus clustering, essentially treating each cluster label as a 'vote' then reassigning final cluster membership based on total votes across all cluster structures (for more detailed explanation please see references) [100, 101]. The term has also been used to describe the human-machine interaction required to qualitatively assess if the underlying concepts communicated in different clustering structures are consistent with each other [100].

#### 3. Software and machine learning model parameters

Analysis was undertaken in Stata version 15 (Stata Corporation, College Station, TX, USA) Python and R [102, 103]. The scikit-learn package in Python [90] was used for the PCA and clustering algorithms and the Keras package [104] was used to rank feature importance.

#### 4. Resources and data governance

Data access was subject to the conditions for use of Child and Youth Development Research Partnership (CYDRP) data repository. All project data was stored and accessed from the secure CYDRP data server in keeping with the CYDRP data security declaration and the conditions of research ethics approval in which the student is a named investigator (HREC 2016–2708). This project was approved by the CYDRP Steering Committee and reviewed by the CYDRP First Nations Advisory Group who approved the methodology, aims and objectives of the study.

## Results

### Initial descriptive analysis

Table 1 contains descriptive statistics of our study cohort (n = 8,267). There was a distinct difference in remoteness between Aboriginal and non-Aboriginal children. The majority of Aboriginal children were from remote regions (their mothers reported place of residence at the time of their birth) while the majority of non-Aboriginal children were from urban regions. There were higher rates of young maternal age, premature birth, maternal alcohol use and smoking in pregnancy in Aboriginal compared to non-Aboriginal children. There were higher rates of hospitalisation and child protection contact amongst Aboriginal children than non-Aboriginal children (Table 1). In terms of school experience, Aboriginal children experienced higher school mobility and lower school attendance (Table 1).

### Clustering

Three variable sets were used to create three sets of clusters, as described in the methods. Table 2 below summarises the models, including the number of clusters, size and descriptive names for the clusters. For the list of the features used in methods 1 to 3 please see S4 Appendix.

**Table 1. Characteristics (%) of our study cohort born in the NT from 2006 to 2009.**

| Characteristics | Aboriginal (n = 4,624) | non-Aboriginal (n = 3,643) |
|---|---|---|
| **Sex** | | |
| Male | 48.0 | 49.6 |
| Female | 52.0 | 50.4 |
| **Remoteness** | | |
| Urban | 35.0 | 77.7 |
| Remote | 65.0 | 22.3 |
| **Perinatal factors** | | |
| Maternal anemia in pregnancy | 4.9 | 0.9 |
| Gestational diabetes | 10.8 | 6.9 |
| Maternal STI in pregnancy | 9.0 | 1.0 |
| <7 antenatal care visits attended | 37.4 | 16.0 |
| Mother <18 years old at delivery | 11.1 | 1.2 |
| First born child | 30.6 | 41.2 |
| Preterm birth | 12.7 | 6.9 |
| Low birth weight | 12.3 | 6.0 |
| APGAR<7 | 2.6 | 1.3 |
| Neonatal resuscitation required | 54.0 | 52.0 |
| Maternal alcohol use in pregnancy* | 13.3 | 6.3 |
| Maternal smoking in pregnancy** | 50.7 | 18.9 |
| **Hospital diagnosis from birth to age 5** | | |
| Avoidable hospitalisation | 59.7 | 19.9 |
| Infancy related condition | 32.6 | 18.1 |
| Gastrointestinal disease | 29.0 | 5.1 |
| Skin infection | 21.3 | 2.0 |
| Nutritional deficiency | 20.7 | 0.7 |
| Acute bronchiolitis | 18.6 | 3.4 |
| Injury | 17.0 | 8.0 |
| Bacterial pneumonia | 16.1 | 2.7 |
| Otitis media | 16.0 | 3.1 |
| Acute URTI^ excluding croup | 12.9 | 5.0 |
| Dental condition | 7.1 | 1.7 |
| **Child Protection contact from birth to age 5** | | |
| Any notification | 46.9 | 12.3 |
| Any substantiation | 23.0 | 2.9 |
| Notification by police | 25.4 | 5.4 |
| Notification by health professional | 21.2 | 2.2 |
| Notification by school personnel | 4.3 | 1.9 |
| Substantiated attributed to DV | 13.2 | 1.8 |
| Substantiation attributed to AOD | 9.0 | 0.9 |
| **Year 1 school experience** | | |
| Attending more than one school | 29.2 | 4.5 |
| School attendance rate (mean) | 71 | 93 |

*744 missing (Aboriginal children), 236 missing (non-Aboriginal children)

**856 missing (Aboriginal children), 386 missing (non-Aboriginal children)

^Upper Respiratory Tract Infection

**Table 2. Result for three clustering methods.**

| Cluster method | Number of clusters, K | Cluster names | Proportion of cohort in each cluster (%) |
|---|---|---|---|
| Model 1: Expert selection | 5 | Non-attenders | 20 |
| | | Thriving | 65 |
| | | Neglect | 4 |
| | | Abuse | 7 |
| | | Ill | 4 |
| Model 2: PCA | 4 | Mixed, low vulnerability | 14 |
| | | Thriving | 79 |
| | | Mixed, high vulnerability | 4 |
| | | Ill | 4 |
| Model 3: Post cluster variable extraction | 5 | Non-attenders | 12 |
| | | Abuse | 3 |
| | | Thriving | 77 |
| | | Neglect | 2 |
| | | Ill | 7 |

## Post-hoc analysis of clusters

As evident in Table 2, each of the three methods separated the dataset into subpopulations which demonstrated that the most vulnerable clusters experience overlapping social, health and education risks. All methods estimated that the most vulnerable children comprise 10–15% of the population and separated these children from the low-risk clusters. The post-hoc analysis was performed using the 17 key variables identified in method 3 of dimensionality reduction, as these were the most informative and interpretable of the original 43 variables. We present abbreviated results of Model 1 below, with full post-hoc analysis of Model 1, Model 2 and Model 3, found in S5 Appendix.

The five clusters identified in Model 1 were named *neglect*, *abuse*, *ill*, *thriving* and *non-attenders*. A descriptive summary, across the four clustering variables and four key demographic characteristics, is presented in Table 3. Two of these factors were included as they are significant for the NT context–the proportion of each cluster that was Aboriginal and the

**Table 3. Post-hoc cluster analysis of Model 1.**

| Cluster | Non-attender | Thriving | Neglect | Abuse | Ill |
|---|---|---|---|---|---|
| **Cluster size** | | | | | |
| number of children in each cluster (n) | 1639 | 5361 | 343 | 594 | 330 |
| proportion of each group in study cohort (%) | 20 | 65 | 4 | 7 | 4 |
| **Median** | | | | | |
| Year 1 attendance rate | 46 | 92 | 74 | 84 | 60 |
| Hospitalisations by 5 | 2 | 0 | 3 | 2 | 9 |
| Neglect notifications by 5 | 0 | 0 | 4 | 1 | 1 |
| Abuse notifications by 5 | 0 | 0 | 1 | 3 | 0 |
| **Proportion (%)** | | | | | |
| Aboriginal | 97 | 36 | 91 | 79 | 94 |
| Remote | 68 | 18 | 41 | 26 | 62 |
| Born premature | 13 | 8 | 16 | 12 | 26 |
| Alcohol in pregnancy | 10 | 7 | 33 | 18 | 19 |

proportion from a remote area (meaning that their mothers lived in a remote location at time of birth). Two perinatal factors were also included–maternal alcohol use in pregnancy and prematurity, as these factors have known links to childhood ill health and neglect.

The '*neglect*' group contained 343 (4%) of children. In this group, the median number of notifications for neglect was 4, median school attendance was 74%, compared to the national average of 93% for Year 1 school attendance in 2018 [105]. Children in this group had high hospital admissions, with median of 3. Although primarily experiencing neglect, there was also evidence of risk of abuse, with a median number of 1 abuse notification.

The '*abuse*' group contained 594 (7%) children. Median notifications for abuse were 3, with median school attendance of 84%–higher than the neglect group. This group had a median of 2 hospitalisations. The abuse group had some risk of co-existing neglect, with median of 1 neglect notification by age five.

The '*ill*' group consisted of 330 (4%) children, with a median of 9 hospital admissions. This group had a median of 1 neglect and 0 abuse notifications. This group had the lowest school attendance–a median of 60%.

The '*thriving*' group was the largest group, with 5361 (0.65) children. This group had a median of 0 abuse or neglect notifications, and a median of 0 hospital admissions. The median school attendance was 91.45%.

Finally, the '*non-attending*' cluster had low school attendance, of 46%. The non-attending cluster had a moderate number of hospitalisations, with a median of 2 and a median of 0 neglect or abuse notifications.

The '*non-attending*', '*neglect*' and '*ill*' clusters were predominantly comprised of Aboriginal children (91–97%), and a substantial proportion were from remote areas (41–68%). The '*ill*' group had the highest proportion of premature birth (26%) and the '*neglect*' group had the highest proportion of maternal alcohol use in pregnancy (33%).

## Discussion

### Pattern of vulnerability across the domains of education, health and child protection

Clustering NT children based on their health, education and child protection data consistently identified a group of highly vulnerable children who comprise up to 15% of the cohort, and experience overlapping risks of poor health, low school attendance and contact with the child protection system. Models 1 and 3 identified *neglect*, *abuse*, *ill*, *thriving* and *non-attenders* clusters, whereas Model 2 identified *ill*, *thriving*, *mixed low- risk* and *mixed high-risk* clusters, rather than separating them out by *type* of vulnerability.

The segregation of vulnerable children by risk type in Model 1 and Model 3 is considered to be more explanatory of the data and is consistent with previous CYDRP research–specifically the finding that, in the NT, a subset of children only receive reports for neglect or only receive reports for emotional abuse, while a small minority receive overlapping reports across all types of abuse–presumably representing different subpopulations [81]. The implication of this finding is that, depending on the type of abuse notification, children may come from different risk clusters and require different interventions. Whilst out-of-home care is avoided where possible it is sometimes necessary to protect children from immediate harm from violent and dangerous homes and removal may be required for some of the 7% of children identified as belonging to the *abuse* cluster [106]. On the other hand, children experiencing neglect may benefit from support through family and parenting interventions, a more suitable response for the 4% identified in the *neglect* clusters by this study [107].

The *ill* group had the highest rate of premature birth, which is a known risk factor for a variety of later childhood illnesses, particularly lung disease [108, 109]. This may have contributed to the ill health of some members of this cluster, however, the *neglect* and *ill* groups particularly overlapped. The 4% of children in the *ill* cluster tended to have some contact with CPS via neglect notifications, and the 4% of children in the *neglect* cluster also tended to have higher than average hospitalisations. As discussed earlier, this likely represents a two-way relationship with children experiencing neglect being more likely to become unwell–possibly due to poorer hygiene, diet, reduced preventative health and reduced parental supervision (injury) [70]. Conversely, non-neglected children who have frequent hospital visits have higher exposure to health professionals, a primary reporter source for neglect notifications [81]. Of the highly vulnerable clusters, the *neglect* and *ill* clusters have lower median school attendance than the abuse cluster. School absenteeism has been found to be associated with any contact with the child protection system, however with a particular association with neglect substantiations [4]. This has been hypothesised as a direct effect of reduced parental supervision [4]. The *neglect* cluster also had the highest proportion of maternal alcohol use in pregnancy. This may be for two reasons. Firstly, risky alcohol use in pregnancy triggers an automatic mandatory report of neglect to the CPS at the time of birth. Secondly, previous research has found that children of mothers with alcohol use disorders are at higher risk of later contact with the CPS, particularly with neglect [110, 111].

As discussed below, these clusters with overlapping risks of ill health and neglect are primarily from remote areas (unlike the abuse cluster which is primarily urban) and are 94% Aboriginal. The remote families these clusters represent are potentially a target for evidence based remote parenting support programs that incorporate cultural elements of early child rearing significant to Aboriginal people [112, 113]. There is some evidence that a shortage of services through remote health clinics could contribute to increased hospitalisation [114] but not to the extent to fully explain this cluster, Rather than simply increasing health services, the combined risks of neglect and ill health may be better targeted via parenting interventions.

Whilst the cluster algorithms generally separate children with high rates of hospitalisations, abuse or neglect, they do overlap. This provides support for the concept of holistic, 'all-of-child' programs, in the early years of life when the pattern of neglect or abuse is not clear. In these high-risk groups, the increased hospitalisation and notifications are present from the first few months of life, and could trigger a holistic, rather than a single system target intervention (i.e. just health or just CPS). Nurse home visits are one example of a holistic intervention and reduce later involvement with the child protection system [115], and have been implemented in parts of the NT [116]. Given these three clusters *combined* comprise less than 15% of the cohort, it may be feasible to aim for progressive universalism, with this group specifically targeted by effective, resource intensive interventions such as nurse home visits.

The highly vulnerable groups differed geographically depending on type of risk–the *ill* group was majority remote, the *neglect* group nearly half remote, whereas the *abuse* group was 74% urban. As discussed previously, exposure to domestic violence is considered a form of emotional abuse and leads to a mandatory report from police, if they attend an incident [86]. An analysis of >80,000 cases of intimate partner violence in the NT (2009–2014) found higher incidence in urban centres [117]. In remote areas, there were an average of 202 incidents per 1000 population, compared to average of 782 per 1000 population in urban areas [117]. Furthermore, a retrospective analysis of data specific to Royal Darwin Hospital (the sole trauma referral centre in the NT), from the Australia New Zealand Trauma Registry found that in the NT, being injured from intimate partner violence in an urban or remote, as opposed to very remote, location carried higher odds of previous presentations with intimate partner violence [118].

## School attendance rate and vulnerability

To interpret the significance of school attendance in defining the clusters, we have used the concept of evidence accumulation [100] to extract the most significant and consistent insights. Firstly, all three models identified a clearly thriving group, with low hospitalizations and CPS contact. Notably, for Model 2 and Model 3, some children in the *thriving* cluster had low school attendance, suggesting that children can be otherwise thriving, but with poor school attendance. In Model 1, where no children with poor attendance existed in the *thriving* cluster, the *non-attenders* cluster closely mirrors the thriving group in terms of low CPS and health risks, but has lower attendance.

Given these thriving non-attenders lack the risks traditionally associated with poor attendance (neglect, abuse or poor health), poor attendance may be more a reflection on the suboptimal availability of western school options. There has traditionally been a deficit framing surrounding the education of Aboriginal children growing up in remote communities [119]. The fact that some children with poor attendance are thriving in other ways suggests that they are not in deficit, but are perhaps learning a different set of skills and values from those traditionally taught and measured in western education [120]. A similar comment was made after the 2015 introduction of Direct Instruction as a new curriculum for remote NT schools. This has been described as a prescriptive teaching methodology, originally designed to bring developmentally delayed American children up to a minimum standard on a specific set of skills, and mis-applied to remote NT schools, who have majority Aboriginal, multilingual and not developmentally delayed students. It was suggested that poor attendance might be a consequence of a program that bored students and disenchanted teachers [121]. The contentious recommendations to close remote high schools [122] are similarly concerning, driven by a desire for statistical parity among Aboriginal and non-Aboriginal students, but disregarding the documented emotional toll caused by dislocation of students from their families and communities, and its association with past policies of forced assimilation [123, 124]. The NT Government funding of schools based on attendance was likely to further exacerbate these issues, as poorly attended schools that most need support are then least able to access it–and without quality education options available at homelands/outstations, many students dis-engage from schools [125]. Fortunately, NT Government funding of schools is under review and is proposed to shift to enrolment based funding [126].

The *non-attenders* cluster may benefit from education programs which inspire them to engage, and from attendance programs that are strengths-based and community embedded. An example of valuing Aboriginal cultural knowledge in education is the 'two way' curricula, which embed Aboriginal and Western knowledge into the formal curriculum–i.e. as promoted by the 'Growing Our Own' program of teacher education [127, 128]. Strengths-based attendance programs include the Clontarf Foundation, which aims to improve education, self-esteem and life skills for young Aboriginal and Torres Strait Islander men, and programs such as the Stronger Smarter Sisters program, implemented at Katherine High School [129, 130]. Finally, the Remote School Attendance Strategy is a program running in 83 schools across Australia, which employs community members to help improve children's school attendance. The nation-wide impact of the Remote School Attendance Strategy appears to be modest but has been embraced by some communities [131–133].

## Limitations

Health data was collected for administrative purposes and therefore does not provide granularity or textual notes to fully explain admissions. Furthermore, health data is restricted to hospital admissions and does not capture other health service events including clinical assessments

in remote health clinics, a key component of health care in the remote NT. Secondly, our key education variable was attendance in Year 1, which is a coarser measure of developmental vulnerability than a purpose designed measurement, such as the AEDC. Future work may investigate how the 'non-attending but thriving' cluster scored on the AEDC. Thirdly, notifications or substantiations recorded in the CPS data are only a proxy for abuse and neglect of a child and may not capture every case. Also, our study was likely to underestimate the domestic violence rates as only domestic violence substantiated notifications were included in the study due to data limitations.

The completeness of reporting of events is also an important factor when considering the *non-attenders* cluster, as data collection may be limited by service availability if the regions with poorer school attendance also have a more general limit on services including fewer potential CPS reporters. It has been suggested that intimate partner violence, for instance, may be under-reported in very remote settings [118]. This is less likely in the case of child maltreatment, firstly given NT mandatory reporting laws require every person in the NT to report child abuse or neglect, and secondly given the transparency provided by communal living in remote communities, it is likely that most maltreated children will be recorded, even if not every event [86].

A further limitation is the absence of methods to establish the best fit of the three cluster structures. There are no objective methods to determine the most important or informative of our clustering results. Therefore, there may be other ways to interpret the patterns existing in the data, which are not shown in our exemplary model [94]. To account for this limitation our analysis drew on the concept of evidence accumulation [100]. In this study, each of the clustering methodologies provided evidence that there was a minority group of children who experienced complex, overlapping risks and were differentiated primarily based on maltreatment type and hospitalisation level. The incidentally noted intermediate risk group, the *non-attenders* were present in only some cluster methodologies, and should therefore be interpreted with more caution.

This project provides evidence to support the development of cross agency, collaborative interventions in early childhood. This analysis secondly identified that poor school attendance, particularly in remote areas, may be independent of other early childhood risks, which may support the concept of providing interventions informed by the pre-existing strengths of children in remote communities. Whilst the majority of the discussion has a focus on Aboriginal children, a substantial minority (up to 20%) of the highest risk groups are non-Aboriginal children who also require effective early childhood interventions.

## Conclusion

The use of unsupervised machine learning techniques on a large, linked health, education and child protection dataset, of 8267 children from the Northern Territory of Australia, produces clusters that describe differing patterns of risk. Results from each of the three clustering methods found that 10–15% of children are very high risk, and this is differentiated into children who are predominantly ill, experience neglect or experience abuse, and all high risk groups have low school attendance. These highest risk groups experience vulnerability across all three domains, supporting the need for early, holistic intervention before more specific approaches may become necessary as these groups differentiate later in childhood. Interagency cooperation is central to delivering a suitably collective and coordinated response for the most vulnerable children. A secondary finding was of a large group of, predominantly Aboriginal, children who have poor school attendance in their early years but are otherwise thriving and may be a target for strengths-based remote school attendance programs.

## Supporting information

**S1 Appendix. Rationale for inclusion of CPS variables into models** [14, 18, 20, 58, 134].
(DOCX)

**S2 Appendix. Details on description of K-means** [89, 135].
(DOCX)

**S3 Appendix. Detail on dimensionality reduction methods** [91, 93–97].
(DOCX)

**S4 Appendix. Detail on the 3 dimensionality reduction methods** [58, 90, 93, 94, 98, 99, 136–140].
(DOCX)

**S5 Appendix. Full model results.**
(DOCX)

## Author Contributions

**Conceptualization:** Steven Guthridge.

**Data curation:** Vincent Yaofeng He, Steven Guthridge.

**Formal analysis:** Lucinda Roper, Vincent Yaofeng He.

**Funding acquisition:** Steven Guthridge.

**Investigation:** Lucinda Roper, Steven Guthridge.

**Methodology:** Lucinda Roper, Vincent Yaofeng He, Oscar Perez-Concha.

**Project administration:** Steven Guthridge.

**Software:** Lucinda Roper, Vincent Yaofeng He.

**Supervision:** Steven Guthridge.

**Visualization:** Lucinda Roper.

**Writing – original draft:** Lucinda Roper.

**Writing – review & editing:** Lucinda Roper, Vincent Yaofeng He, Oscar Perez-Concha, Steven Guthridge.

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
