## [Decision Letter · Decision Letter 0]

1 Jul 2022

PONE-D-22-12337 Complex early childhood experiences: characteristics of Northern Territory children across health, education and child protection data PLOS ONE

Dear Dr. Roper,

Thank you for submitting your manuscript to PLOS ONE. After careful consideration, we feel that it has merit but does not fully meet PLOS ONE’s publication criteria as it currently stands. Therefore, we invite you to submit a revised version of the manuscript that addresses the points raised during the review process. The comments provided by the individual reviewers - sent with this writing - note the potential (impact) of the manuscript, but at the same time several aspects are identified which require additional attention. Based on my own additional reading, I concur in these evaluations. In line with the reviewer comments, the current manuscript shows a strong and in-depth methodological approach, however grounding in a theoretical framework is underrepresented in the introduction. A more balanced approach is advised. Please try to address the reviewer concerns through your best efforts. Although all comments require attention, of special concern are 1) the abstract, 1) overall grammar and punctuation, 2) the clarity in the abstract, 3) the discrepancies in sample size(s), 4) the issue raised by reviewer 2 on the input of Native Aboriginals in the interpretation and message brought through this manuscript and 5) the framing of the current work in a theoretical framework or model. In addition, I have some concerns related to the K-means approach, listed below under Editor Comments. Please submit your revised manuscript by Aug 15 2022 11:59PM. If you will need more time than this to complete your revisions, please reply to this message or contact the journal office at plosone@plos.org. Please include the following items when submitting your revised manuscript:A rebuttal letter that responds to each point raised by the academic editor and reviewer(s). You should upload this letter as a separate file labeled 'Response to Reviewers'.A marked-up copy of your manuscript that highlights changes made to the original version. You should upload this as a separate file labeled 'Revised Manuscript with Track Changes'.An unmarked version of your revised paper without tracked changes. You should upload this as a separate file labeled 'Manuscript'.

We look forward to receiving your revised manuscript.

Kind regards,

Ralph C. A. Rippe, Ph.D.

Academic Editor

PLOS ONE

Journal Requirements:

4. Please upload a new copy of Figure 3 as the detail is not clear. Please follow the link for more information: https://blogs.plos.org/plos/2019/06/looking-good-tips-for-creating-your-plos-figures-graphics/" https://blogs.plos.org/plos/2019/06/looking-good-tips-for-creating-your-plos-figures-graphics/

**Additional Editor Comments**

E1) the authors write that K-means is a simple technique (and I agree), but use a very complex statistic (silhouette) for evaluation. Although useful, it is difficult to grasp for non-involved readers. The manuscript does not describe why statistics well-known to more readers (such as the CH-index or DB-index) were not used (in addition to the silhouette) and how they (do not) relate.  E2) the authors rely on K-means before and after dimensionality reduction, but do not reflect on the fact that K-means is also not very strong at dealing with different cluster densities and cluster shapes, and do not allow for cluster boundary overlap. Could the authors reflect on the implications of this choice over e.g. density mixtures, especially in higher dimensionality in the current application? E3) the authors describe that they applied 10 (random?) starts to circumvent the local minimum issue. However, with higher K, the number of local minima strongly increases, and thus a higher number of random starts than 10 is needed. A plethora of examples shows that local minima are still encountered with 100+ random starts for K=6. If the authors did not use random starts, the exact initialization procedure should be explained.

Reviewers' comments:

Reviewer's Responses to Questions

**Comments to the Author**

1. Is the manuscript technically sound, and do the data support the conclusions?

Reviewer #1: Yes

Reviewer #2: Partly

2. Has the statistical analysis been performed appropriately and rigorously? 

Reviewer #1: I Don't Know

Reviewer #2: I Don't Know

3. Have the authors made all data underlying the findings in their manuscript fully available?

Reviewer #1: Yes

Reviewer #2: Yes

4. Is the manuscript presented in an intelligible fashion and written in standard English?

Reviewer #1: Yes

Reviewer #2: Yes

5. Review Comments to the Author

Reviewer #1: Thank you for the opportunity to review the manuscript describing important research that links databases across service sectors (education, health, and child welfare) in the Northern Territories in Australia. My overall concerns with the manuscript are that it needs to be revised for clarity and that potential biases in child abuse and neglect reporting are not emphasized enough. Please see below for more detailed comments and suggestions:

OVERALL

1) The manuscript should needs some revision related to grammar and punctuation. For example, several places have unnecessary commas and 'data' should be plural not singular, i.e. data are rather than data is...

ABSTRACT:

2) The first sentence of the abstract is so non-specific that it isn't particularly helpful. Please provide more specificity or delete that sentence. In general, the first paragraph could be shortened to provide additional information as I have requested below in the limited space.

3) The research question or objective should be more clearly stated in the abstract.

4) The abstract should clearly identify the study type (ie retrospective cohort) and how the research cohort was identified.

5) The abstract should clearly identify the specific data sources that were used beyond the broad description that is currenlty provided.

INTRODUCTION:

6) Given the complexity of the research project, the manuscript and project would be improved if they were clearly guided by a theoretical or conceptual model. For example, this project and resulting manuscript could be framed using the social-ecological model. The introduction would be clarified if it were more structured using the social-ecological model or another model that the authors think is most appropriate. Then, the structure that was set up in the introduction could be carried on in the results and the discussion.

METHODS:

7) The section describing how data were linked at the individual level then de-identified (I think) was unclear and should be revised for clarity.

8) The method for identifying the study cohort should be clarified. Consider using a flow diagram that shows the different data sources and how those were integrated to create the cohort that was ultimately used for the analysis.

METHODS/RESULTS:

9) Related to my comments (#7 and #8 above) The cohort of 8,267 children seems low given that 15,284 were born during the selection period--the paper should clearly indicate that 8,267 represents 54% of those born during the selection period. If 65% of all Australian children and 83% of Aboriginal children attended government schools, shouldn't the study cohort be closer to at least 65%? What happened to the 'missing' 10 to 20%?

DISCUSSION:

10) As noted above, a theoretical or conceptual model should be used to structure the discussion.

11) While CPS reporting bias is included in the limitations, I think the potential for this bias should be reported and discussed earlier and in more detail. Systemic oppression is a cause of poverty and diminished access to resource as well as a cause of toxic stress all of which can lead to an actual increased incidence of child abuse and neglect. However, the authors should note more strongly that systemic racism may also influence reporting and substantiation decision-making.

12) I agree with the authors that cross-sector interventions should be considered; however, can they provide more examples or what these types of interventions might look like?

Reviewer #2: PlosONE-D-22-12337

Cluster analysis of dataset including 8k NT children now 13 to 16y. The premise is a very important one that a subset of children have a multitude of challenges (whereas most studies adjust for the association of one factor against another rather than accept both as cumulative risks). This is interesting work but I have a few comments and suggestions for the authors.

Abstract:

• Suggest including more detail on datasets and summary of how clustering was done- I understand this is complex but a summary of L443 paragraph would help the reader follow what was done and thereby interpret the findings. There are many ways to determine clustering and what you accept as a clear pattern of similarity

• Child Protection Services should be expanded as different anagrams in different jurisdictions

• Suggest explaining/defining what you mean by ‘moderate’ and ‘mostly’ Aboriginal

• It would be good for the reader to have a summary of the dataset (e.g., description of the variables in line 245)

Background:

• L40 suggest rewording society ‘should’ support optimal development- not sure we do!

• L58 suggest replace ‘developed countries’ with ‘high income countries’

• L70-79, 11: data should be plural throughout

Methods

• What is the potential effect of excluding 7k children with no records on attendance?

• L255: 5 years

• Not sure I am convinced about using the 4 variable set described. Hospitalisations and attendance rates are outcomes whereas number of neglect and abuse notifications are risk factors from my perspective. Also, I completely understand why you might not disaggregate by ethnic variables in other settings, but in the Northern Territory, there is a substantial difference between the lived experience of a remote Aboriginal child and an urban non-Aboriginal child. Can you explain the justification for not running two analyses here (one analysis for clustering in Aboriginal kids and a completely separate cluster analysis for non-Aboriginal) pls? Although you have explained the justification for doing this work to inform decision makers about coordinated services, this would be very different for Aboriginal (cultural safety, and Aboriginal led design) vs. non-Aboriginal. I think the reason for not doing this needs much more detailed explanation at least as you have found 3 clusters that are almost all Aboriginal (non-attender, neglect and ill) and 2 clusters with lower rates of Aboriginal kids.

Results

• L415-421: all descriptions of ‘higher’ should have p values or confidence intervals

• Table 1 footnote: does 744 missing (Aboriginal children) mean there were 980 missing values of which 744 were Aboriginal and 236 not? It just seems unusual to word this way and does that mean there were no missing data for other variables except the 2 with asterisks? That seems unusual so it might be more informative to have n/Ns too?

• Suggest 5 years rather than 5

• L464 has the ill group with the ‘lowest school attendance’ whereas L471 is lower

• I am afraid I cannot really read the Figure at the end, and it is confusing as it appears to suggest that 100% of the ‘thriving’ category have attendance <80%, whereas only 15% of the ‘abuse’ cluster has attendance <80%

• In several places, including text and appendices, proportions are presented with whole percentages in one place and then 2 decimal places in another. Suggest could be all rounded to whole numbers for this sort of study.

Discussion

• L628: Not sure I agree with the comment that you are unlikely to have differential misclassification of maltreatment. Indeed, it is almost certainly the case that Aboriginal communities are much less likely to report any form of abuse given the long history of the Stolen Generations in Australia with community fear of removal if there is anything adverse found in relation to a child (also a reason for less attendance at healthcare)

Finally, these issues are quite complex and potentially sensitive. In Aboriginal research there is a move to ensure an Aboriginal perspective. I do not know the authors, but I think it would be critical to have Aboriginal authors (more than one) or Aboriginal reviewers of this work to assist with ensuring that it is not sending the wrong message that the problems only exist among Aboriginal children. The dataset was mostly Aboriginal kids (4,624 vs 3,643), whereas in the Territory you state only 43% of children are Aboriginal, so any cluster in your dataset is automatically more likely to have a higher proportion of Aboriginal children than otherwise and further reason to disaggregate the clustering by Aboriginality so that the findings can be targeted appropriately if the clusters are different in the 2 populations.

6. PLOS authors have the option to publish the peer review history of their article (what does this mean?). If published, this will include your full peer review and any attached files.

Reviewer #1: **Yes: **Mandy A Allison

Reviewer #2: No

---

## [Author Response · Author response to Decision Letter 0]

9 Dec 2022

25/November/2022

Associate Professor Ralph C. A. Rippe, 

Academic Editor,

PLOS ONE

Dear Associate Professor Rippe,

RE: PONE-D- 22-12337

Complex early childhood experiences: characteristics of Northern Territory children across health, education and child protection data

We thank the editor and reviewers for their comments. Their insights and suggestions have assisted the authors greatly. We have considered each comment and provide the following responses, along with the proposed revisions to the manuscript. We have tracked the proposed changes in the revised manuscript.

Editor’s comments

We have ensured that our manuscript meets PLOS ONE's style requirements.

Thanks. We provide contact information for a data access committee, ethics committee, or other institutional body to which data requests may be sent.

Thanks, previously, we have added the followings:

The study datasets contain sensitive personal information and are held on a secure cloud-based server with restricted access. Access requires the approval of the ethics committee and data custodians. For applications for data access, please contact the Menzies Data-linkage Program Leader at steve.guthridge@menzies.edu.au. 

4. Please upload a new copy of Figure 3 as the detail is not clear. Please follow the link for more information: https://blogs.plos.org/plos/2019/06/looking-good-tips-for-creating-your-plos-figures-graphics/" https://blogs.plos.org/plos/2019/06/looking-good-tips-for-creating-your-plos-figures-graphics/

We apologize for this. To improve the readability, we have included the information as numbers in Table 3, and thus removed the Figure 3 from the manuscript (similar response to Reviewer 2 Comment 14). As such, a revised Figure is no longer relevant.

Additional Editor Comments

E1) the authors write that K-means is a simple technique (and I agree), but use a very complex statistic (silhouette) for evaluation. Although useful, it is difficult to grasp for non-involved readers. The manuscript does not describe why statistics well-known to more readers (such as the CH-index or DB-index) were not used (in addition to the silhouette) and how they (do not) relate. 

Thank you for your comment. We agree that other scores, such as Calinski-Harabasz (CH-index) and Davies-Bouldin (DB-index), could have been used. Nevertheless, the three scores have a similar meaning, that is, they return a higher value when clusters are well separated and dense, which relates to a standard concept of a cluster. 

In addition to this, we used the Silhouette Coefficient because:

1. This score is bounded between +1 for highly dense clustering and -1 for the opposite. Scores around zero indicate overlapping clusters. 

2. This score is commonly used in the machine learning community and we are very familiar with it. 

E2) the authors rely on K-means before and after dimensionality reduction, but do not reflect on the fact that K-means is also not very strong at dealing with different cluster densities and cluster shapes, and do not allow for cluster boundary overlap. Could the authors reflect on the implications of this choice over e.g. density mixtures, especially in higher dimensionality in the current application?

We carried out experiments using Gaussian Mixture Models (GMM) in parallel to K-means. The clusters yielded from GMM provided similar insights into our research question. The inclusion of both K-means and GMM added considerable length and complexity to the paper, so we therefore decided not to show GMM results.

E3) the authors describe that they applied 10 (random?) starts to circumvent the local minimum issue. However, with higher K, the number of local minima strongly increases, and thus a higher number of random starts than 10 is needed. A plethora of examples shows that local minima are still encountered with 100+ random starts for K=6. If the authors did not use random starts, the exact initialization procedure should be explained.

Thank you for your comment. We didn’t use a random initialisation.. We used the “k-means++” initialisation method that the scikit-learn Python library provides (line 684). “k-means++’ : selects initial cluster centroids using sampling based on an empirical probability distribution of the points’ contribution to the overall inertia. This technique speeds up convergence, and is theoretically proven to O(logk) be -optimal.” (Source: https://scikit-learn.org/stable/modules/generated/sklearn.cluster.KMeans.html)

“K-means ++” is considered a better initialization method than random selection of the initial centroids, needing less start repeats to avoid a local minimum. 

We have added the following explanation in our method section: 

“We used the scikit-learn embedded k-means++ initialisation method for choosing the initial centroids of the clusters. This utilises the furthest-point heuristic to choose points distant from each other and to reduce the chance of convergence on a local minimum.” 

We deleted the sentence that said “several clusters centres are randomly chosen" as this statement was wrong. 

Reviewer 1

Reviewer #1: Thank you for the opportunity to review the manuscript describing important research that links databases across service sectors (education, health, and child welfare) in the Northern Territories in Australia. My overall concerns with the manuscript are that it needs to be revised for clarity and that potential biases in child abuse and neglect reporting are not emphasized enough. Please see below for more detailed comments and suggestions:

We thank the reviewer for her comments. We have addressed her concerns as below.

OVERALL

1) The manuscript should needs some revision related to grammar and punctuation. For example, several places have unnecessary commas and 'data' should be plural not singular, i.e. data are rather than data is...

We thank the reviewer for her suggestions and apologise for our oversight. We have made the revision related to grammar and punctuation.

ABSTRACT:

2) The first sentence of the abstract is so non-specific that it isn't particularly helpful. Please provide more specificity or delete that sentence. In general, the first paragraph could be shortened to provide additional information as I have requested below in the limited space.

Agreed. We have deleted the sentence.

3) The research question or objective should be more clearly stated in the abstract.

Agreed. We have stated the research objective more clearly in the abstract. 

“This is a retrospective cohort study aimed at identifying different groups of children with varying patterns of contact with the child protection, health and education system in the Northern Territory (NT) of Australia.”

4) The abstract should clearly identify the study type (ie retrospective cohort) and how the research cohort was identified.

Agreed. We have added more information in the abstract. Please see response to Comment 3 and the added text below:

The NT perinatal data and government school attendance data were used to identify the 8,267 NT-born children (during 2006-2009) with Year 1 school attendance records in NT government schools.

5) The abstract should clearly identify the specific data sources that were used beyond the broad description that is currently provided.

Agreed. Please see Comment 3 and 4 and the added text below:

“By linking the study cohort to the hospital, child protection and school attendance data, this study used unsupervised machine learning techniques to identify clusters of children who experienced different patterns of risk (i.e, hospitalisation, child protection notification and substantiations) in the first five years of life, as well as low school attendance in Year 1).”

INTRODUCTION:

6) Given the complexity of the research project, the manuscript and project would be improved if they were clearly guided by a theoretical or conceptual model. For example, this project and resulting manuscript could be framed using the social-ecological model. The introduction would be clarified if it were more structured using the social-ecological model or another model that the authors think is most appropriate. Then, the structure that was set up in the introduction could be carried on in the results and the discussion.

We thank the reviewer for this feedback, and we have incorporated the framework into our manuscript. We have added the following two paragraphs to the introduction section in our manuscript (L2).

“According to the Bronfenbrenner's Ecological Systems Theory, a child’s development is affected by the surrounding environment, ranging from family to school to society. This bio-ecological framework recognizes that risk exposure accumulates over time and the nature of risk often co-occurs.”

METHODS:

7) The section describing how data were linked at the individual level then de-identified (I think) was unclear and should be revised for clarity.

We have revised the section in page 10 to improve the clarity:

“The four datasets are part of data are held in a large data repository containing that contain de-identified, unit-level linked records for NT children across a total of 14 datasets. The repository is longitudinal dataset,was developed through the Child and Youth Development Research Partnership (CYDRP) between Menzies School of Health Research and multiple NT Government agencies, including the departments of Health, Education, and Territory Families, Housing and Communities (child protection)[84]. Initial linkage was conducted by SA NT Datalink [85], using probabilistic methods to match the records for children across multiple datasets, with clerical review of uncertain matches. This process is confirmed to result in 99.6% accuracy for completed links [85]. SA NT DataLink creates a unique linkage key for each child and provides this to data custodians. Each data custodian then creates a de-identified research dataset containing only the linkage keys and approved research variables, which is provided to researchers. Researchers are then able to merge records for the same child across multiple datasets. A full description is available elsewhere [86].”

8) The method for identifying the study cohort should be clarified. Consider using a flow diagram that shows the different data sources and how those were integrated to create the cohort that was ultimately used for the analysis.

Agreed, we have added the following text

“Figure 1 shows the flow diagram in which the study cohort was selected. The perinatal data was used to identify the 15,284 children born in the NT from 2006-2009. To ensure each child has complete information for Year 1 school attendance and complete history of hospitalisation and contact with the child protection system in the first five years, we linked the perinatal data to the NT Government school enrolment and attendance data, excluding children without records of Year 1 school attendance in an NT government school. After applying the selection criteria, 8,267 children were retained in the study. The 7,017 children who were excluded from our study were children who may have moved interstate or attended NT non-government schools in Year 1; this group of children was mostly non-Aboriginal children (74.5%, n=5,231).”

METHODS/RESULTS:

9) Related to my comments (#7 and #8 above) The cohort of 8,267 children seems low given that 15,284 were born during the selection period--the paper should clearly indicate that 8,267 represents 54% of those born during the selection period. If 65% of all Australian children and 83% of Aboriginal children attended government schools, shouldn't the study cohort be closer to at least 65%? What happened to the 'missing' 10 to 20%?

Issue addressed. Please see our similar response to comment 7.

DISCUSSION:

10) As noted above, a theoretical or conceptual model should be used to structure the discussion.

Agreed. Please see our response to Comment 6 of Reviewer 1.

11) While CPS reporting bias is included in the limitations, I think the potential for this bias should be reported and discussed earlier and in more detail. Systemic oppression is a cause of poverty and diminished access to resource as well as a cause of toxic stress all of which can lead to an actual increased incidence of child abuse and neglect. However, the authors should note more strongly that systemic racism may also influence reporting and substantiation decision-making.

While we agree that systematic racism may influence reporting and substantitation decision-making, our study does not contain evidence to make this claim. As such, we did not discuss systemic racism in our manuscript.

12) I agree with the authors that cross-sector interventions should be considered; however, can they provide more examples or what these types of interventions might look like?

We thank the reviewer. In our discussion, we suggested ‘nurse home visiting program’ might be one holistic intervention that reduces disadvantaged children’s later involvement with the child protection system. In this example, we have cited the study about the Australian Nurse Family Partnership Program for Aboriginal infants and their mothers in Central Australia.

 

Reviewer 2

Cluster analysis of dataset including 8k NT children now 13 to 16y. The premise is a very important one that a subset of children have a multitude of challenges (whereas most studies adjust for the association of one factor against another rather than accept both as cumulative risks). This is interesting work but I have a few comments and suggestions for the authors.

Abstract:

1. Suggest including more detail on datasets and summary of how clustering was done- I understand this is complex but a summary of L443 paragraph would help the reader follow what was done and thereby interpret the findings. There are many ways to determine clustering and what you accept as a clear pattern of similarity.

Agreed. We have included more details. Please see our previous response to Comment 3-5 of Reviewer 1.

2. Child Protection Services should be expanded as different anagrams in different jurisdictions.

Agreed. We have expanded this term. 

3. Suggest explaining/defining what you mean by ‘moderate’ and ‘mostly’ Aboriginal

Agreed. We have included additional information, stating that 97% of the children in the ‘non-attenders’ group were Aboriginal children.

4. It would be good for the reader to have a summary of the dataset (e.g., description of the variables in line 245)

Agreed. We have included more details. Please see our response to Comment 3-5 of Reviewer 1.

Background

5. L40 suggest rewording society ‘should’ support optimal development- not sure we do

Agree, we have replaced “should” with “aim to”, in which we acknowledge it is an aspiration.

6. L58 suggest replace ‘developed countries’ with ‘high income countries’

Agree, we have replaced ‘developed countries’ with ‘high income countries’.

 L70-79, 11: data should be plural throughout

Agree, we have used plural terms for data throughout the manuscript.

Methods

7. What is the potential effect of excluding 7k children with no records on attendance?

We have added the following text in our methods section.

“The 7,017 children who were excluded from our study were children who might have moved interstate or attended NT non-government schools in Year 1; this group of children was mostly non-Aboriginal children (74.5%, n=5,231).”

8. L255: 5 years

Agree, we have changed to ‘5 years’.

9. Not sure I am convinced about using the 4 variable set described. Hospitalisations and attendance rates are outcomes whereas number of neglect and abuse notifications are risk factors from my perspective. Also, I completely understand why you might not disaggregate by ethnic variables in other settings, but in the Northern Territory, there is a substantial difference between the lived experience of a remote Aboriginal child and an urban non-Aboriginal child. Can you explain the justification for not running two analyses here (one analysis for clustering in Aboriginal kids and a completely separate cluster analysis for non-Aboriginal) pls? Although you have explained the justification for doing this work to inform decision makers about coordinated services, this would be very different for Aboriginal (cultural safety, and Aboriginal led design) vs. non-Aboriginal. I think the reason for not doing this needs much more detailed explanation at least as you have found 3 clusters that are almost all Aboriginal (non-attender, neglect and ill) and 2 clusters with lower rates of Aboriginal kids.

Thanks for giving us the chance to clarify the rationale of our study design. We would first like to clarify that our study aim is to identify the different groups of children with varying patterns of contact with child protection, health and education services. Establishing the relationship between risk factors and outcomes is not our study aim. As such, this study did not make prior assumptions about the relationship between different outcomes and Aboriginal status, and as a result, we did not stratify the analysis by Aboriginal status but instead present information about Aboriginal status in the posthoc analysis.

Results

10. L415-421: all descriptions of ‘higher’ should have p values or confidence intervals

The aim of our study is to identify the distinct groups of children with varying patterns of contact with the different child-serving services, and not to compare the difference between Aboriginal and non-Aboriginal children; we therefore did not include this information in our manuscript.

11. Table 1 footnote: does 744 missing (Aboriginal children) mean there were 980 missing values of which 744 were Aboriginal and 236 not? It just seems unusual to word this way and does that mean there were no missing data for other variables except the 2 with asterisks? That seems unusual so it might be more informative to have n/Ns too?

We apologize for the confusion. To improve clarity, we have rephrased the footnote in Table 1 as below. 

“The proportions were derived based on non-missing data. The proportions of missing data in maternal alcohol use in pregnancy for Aboriginal and non-Aboriginal children were 16.1% and 6.5% respectively. The proportions of missing data in maternal smoking in pregnancy for Aboriginal and non-Aboriginal children were 16.1% and 6.5% respectively.”

12. Suggest 5 years rather than 5

Agreed, and we have changed to ‘5 years’.

13. L464 has the ill group with the ‘lowest school attendance’ whereas L471 is lower

We apologize for the typo and corrected the error in the manuscript.

“ The ‘ill’ group… The median school attendance was 60%.”

“Finally, the ‘non-attending’ cluster had the lowest school attendance (46%).”

14. I am afraid I cannot really read the Figure at the end, and it is confusing as it appears to suggest that 100% of the ‘thriving’ category have attendance <80%, whereas only 15% of the ‘abuse’ cluster has attendance <80%

We apologize for this. To improve the readability, we have included the information as numbers in Table 3, and removed the figure from the manuscript.

15. In several places, including text and appendices, proportions are presented with whole percentages in one place and then 2 decimal places in another. Suggest could be all rounded to whole numbers for this sort of study.

We have standardised the way we present results in the manuscript: using whole numbers for median and using one decimal place for proportions.

Discussion

16. L628: Not sure I agree with the comment that you are unlikely to have differential misclassification of maltreatment. Indeed, it is almost certainly the case that Aboriginal communities are much less likely to report any form of abuse given the long history of the Stolen Generations in Australia with community fear of removal if there is anything adverse found in relation to a child (also a reason for less attendance at healthcare).

Response: Agree, and we have deleted the sentence from our manuscript.

17. Finally, these issues are quite complex and potentially sensitive. In Aboriginal research there is a move to ensure an Aboriginal perspective. I do not know the authors, but I think it would be critical to have Aboriginal authors (more than one) or Aboriginal reviewers of this work to assist with ensuring that it is not sending the wrong message that the problems only exist among Aboriginal children. The dataset was mostly Aboriginal kids (4,624 vs 3,643), whereas in the Territory you state only 43% of children are Aboriginal, so any cluster in your dataset is automatically more likely to have a higher proportion of Aboriginal children than otherwise and further reason to disaggregate the clustering by Aboriginality so that the findings can be targeted appropriately if the clusters are different in the 2 populations.

Response: The authors acknowledge the sensitivities and risks for research involving Aboriginal people and are familiar with the recent guidance on working with Aboriginal people including the work by Hartfield and the NHMRC guidelines - Ethical conduct in research with Aboriginal and Torres Strait Islander Peoples and communities: Guidelines for researchers and stakeholders (2018). 

This project fits within a broader body of research – the Child and Youth Development Research Partnership (CYDRP) - which has the specific aim of improving services and outcomes for NT children and young people. Most CYDRP projects report information for Aboriginal children, who are more than 40% of all NT children. The program engages with Aboriginal people at multiple levels. 

• The research is conducted with the approval of a Human Research Ethics Committee, which requires responses by researchers against the NHMRC guidelines for research with Aboriginal people. The HREC review includes review by an Aboriginal research committee who provide advice on projects and have the right of veto for projects.

• All projects within the CYDRP program require approval and ongoing progress reports to the First Nations Advisory Group, who provide guidance on both the relevance and cultural safety of the projects. The advisory group also contributes to the research agenda for the program.

• The results of studies are widely distributed – through scientific papers, research briefs and by presentations, including to relevant Aboriginal service agencies.

---

## [Decision Letter · Decision Letter 1]

5 Jan 2023

Complex early childhood experiences: characteristics of Northern Territory children across health, education and child protection data

PONE-D-22-12337R1

Dear Dr. Roper,

Firstly I would like to apologize for the longer-than-usual procession time of your revised manuscript. The original Reviewer 1 was not available after to evaluate the revised manuscript after all, and has proven difficult to find new reviewers in this stage of the process. Therefore, I have myself again evaluated your responses in light of the original R1.R1 comments, and I come to the same conclusion as that of Reviewer 2 who has also evaluated your revised manscript. We’re pleased to inform you that your manuscript has been judged scientifically suitable for publication and will be formally accepted for publication once it meets all outstanding technical requirements. 

Kind regards,

Ralph C. A. Rippe, Ph.D.

Academic Editor

PLOS ONE

Reviewers' comments:

Reviewer's Responses to Questions

**Comments to the Author**

1. If the authors have adequately addressed your comments raised in a previous round of review and you feel that this manuscript is now acceptable for publication, you may indicate that here to bypass the “Comments to the Author” section, enter your conflict of interest statement in the “Confidential to Editor” section, and submit your "Accept" recommendation.

Reviewer #2: All comments have been addressed

2. Is the manuscript technically sound, and do the data support the conclusions?

Reviewer #2: Yes

3. Has the statistical analysis been performed appropriately and rigorously? 

Reviewer #2: I Don't Know

4. Have the authors made all data underlying the findings in their manuscript fully available?

Reviewer #2: Yes

5. Is the manuscript presented in an intelligible fashion and written in standard English?

Reviewer #2: Yes

6. Review Comments to the Author

Reviewer #2: Thank you for the responses. Although I accept your response in relation to Aboriginal authorship as it is the common standard in Australia, even as a non-Aboriginal researcher, I still feel that we need to move away from the ongoing paradigm of non-Aboriginal researchers conducting studies and just having an Indigenous reference group give the "OK". Aboriginal engagement should be from design to publication and be part of the authorship team if they are helping to design the study and report the findings. Just a comment for future reference from a non-Aboriginal researcher.

7. PLOS authors have the option to publish the peer review history of their article (what does this mean?). If published, this will include your full peer review and any attached files.

Reviewer #2: No

---

## [Editor Report · Acceptance letter]

9 Jan 2023

PONE-D-22-12337R1 

Complex early childhood experiences: characteristics of Northern Territory children across health, education and child protection data 

Dear Dr. Roper:

I'm pleased to inform you that your manuscript has been deemed suitable for publication in PLOS ONE. Congratulations! Your manuscript is now with our production department. 

Kind regards, 

on behalf of

Dr. Ralph C. A. Rippe 

Academic Editor

PLOS ONE